# Secretory Carcinoma of Salivary Gland with High-Grade Histology Arising in Hard Palate: A Case Report

**Kiyofumi Takabatake [1], Keisuke Nakano [1],\*, Hotaka Kawai [1], Saori Yoshida [1], Haruka Omori [1], May Wathone Oo [1], Shan Qiusheng [1], Kenichiro Uchida [2], Katsuaki Mishima [2] and Hitoshi Nagatsuka [1]**

[1] Department of Oral Pathology and Medicine Graduate School of Medicine, Dentistry and Pharmaceutical Sciences, Okayama University, Okayama 7008525, Japan; gmd422094@s.okayama-u.ac.jp (K.T.); de18018@s.okayama-u.ac.jp (H.K.); de20052@s.okayama-u.ac.jp (S.Y.); p4628fuz@s.okayama-u.ac.jp (H.O.); p1qq7mbu@s.okayama-u.ac.jp (M.W.O.); hrbmushanqiusheng@163.com (S.Q.); jin@okayama-u.ac.jp (H.N.)

[2] Department of Oral and Maxillofacial Surgery, Yamaguchi University Graduate School of Medicine, Yamaguchi 7558505, Japan; k.uchida@yamaguchi-u.ac.jp (K.U.); kmishima@yamaguchi-u.ac.jp (K.M.)

\* Correspondence: keisuke1@okayama-u.ac.jp; Tel.: +81-086-235-6651

**Abstract:** Secretory carcinoma (SC) is a recently described salivary gland tumor reported in the fourth edition of World Health Organization classification of head and neck tumors. SC is characterized by strong S-100 protein, mammaglobin, and vimentin immunoexpression, and harbors a t(12;15)(p13;q25) translocation which leads to *ETV6-NTRK3* fusion product. Histologically, SC displays a lobulated growth pattern and is often composed of microcystic, tubular, and solid structures with abundant eosinophilic homogenous or bubbly secretion. SC is generally recognized as low-grade malignancy with low-grade histopathologic features, and metastasis is relatively uncommon. In this case, we described a SC of hard palate that underwent high grade transformation and metastasis to the cervical lymph node in a 54-year-old patient. In addition, this case showed different histological findings between primary lesion and metastasis lesion. Therefore, the diagnosis was confirmed by the presence of ETV6 translocation. Here, we report a case that occurred SC with high-grade transformation in the palate, and a review of the relevant literature is also presented.

**Keywords:** secretory carcinoma; high-grade transformation; *ETV6-NTRK3* fusion; cervical lymph node metastasis

## 1. Introduction

Secretory carcinoma (SC) is a recently recognized malignant tumor arising in the salivary gland. Hirokawa et al. pointed out that the histologic and immunochemical features of salivary gland acinar cell carcinoma resembled secretory carcinoma of the breast [1]. Resis-Fliho et al. noted mammary acinic cell carcinoma found secretory carcinoma to be distinct from acinic cell carcinoma in 2008 [2]. Secretory carcinoma was first documented in salivary glands in a 2010 study through a series of 16 cases [3], and 248 cases have been reported since [4–11]. In the 2017 WHO classification of Head and Neck Tumours 4th ed., SC from mammary analogue secretory carcinoma (MASC) was classified as SC [12].

SC is composed of relatively uniform cells with vesicular nuclei and abundant eosinophilic vacuolated cytoplasm arranged in microcystic, papillary and cystic, solid, macrocystic, and tubular patterns. The tumor has a similar histopathologic morphology to many salivary gland tumors and is particularly difficult to distinguish from zymogen granule-poor acinic cell carcinoma (AciCC).

Immunohistochemically, SC shows GCDFP-15, S-100, and vimentin positive [13,14] mammaglobin which is rarely positive in acinic cell carcinoma. DOG-1 is predominantly negative in SC but usually positive in acinic cell carcinoma, and search of *ETV6-NTRK3* fusion gene is useful for its diagnosis [3,15,16]. In addition, SC generally has been reported as low to medium grade tumor. The high-grade transformation behaves more aggressively and is more associated with a poor prognosis than a low-grade tumor. SC with high-grade histology is rare and only 16 cases have been reported to this date [7,8,11,17–21].

Here we report a case of SC with high-grade transformation in primary lesion and cervical lymph node metastases confirmed with *ETV6-NTRK3* translocation arising in the hard palate of a 54-year-old adult. In this case, the primary lesion was an atypical histological image, and the histological image of the metastatic lesion was different from primary lesion. Therefore, it is a case in which a definitive diagnosis was confirmed with *ETV6-NTRK3* translocation.

## 2. Case Presentation Section

A 54-year-old male was aware of hemorrhage from his hard palate. Physical examination demonstrated a 25 mm × 18 mm, border irregularity, surface granulation tissue-like, ulcerated crater located in the center of his palate at the junction between his hard and soft palate. A head and neck CT showed an irregularly-shaped soft tumor mass of 28 mm× 18 mm on the hard palate, and also showed osteolytic change of the palatal bone, and the tumor infiltrated in the nasal cavity and in the left soft palate. In addition, 25 mm lymph nodes were seen in levels II on the bilateral neck region. There was no clinical or imaging evidence of distant metastasis.

Based on the above results, partial maxillary resection and bilateral neck dissection were performed under clinical diagnosis of primary malignant tumor of the palatal gland and bilateral cervical lymph node metastasis. Informed consent was obtained from the patient.

### 2.1. Pathological Findings of Primary Lesion

Microscopically, the tumor was white and solid mass with relatively clear boundary (Figure 1).

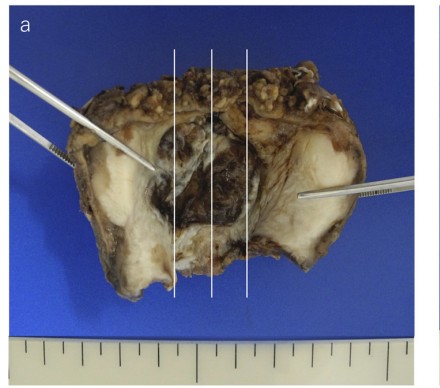 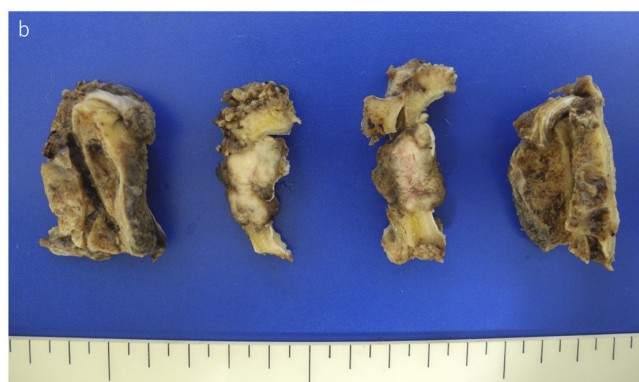

**Figure 1.** Macroscopic view of the tumor. (**a**) The gross resected specimen formed ulcer in hard palate. (**b**) The tumor was a lobulated and white-gray colored tumor.

Histologically, the tumor infiltrated the palatal gland, and destroyed the surrounding existing bone tissue, and the tumor was not encapsulated by fibrous tissue. The tumors exhibited a lobulated growth or papillary growth pattern with narrow fibrous septa (Figure 2a,b). The tumor cells had eosinophilic granular or vacuolated cytoplasm with increasing N/C ratio and with clear nucleoli, however there was no significant number of mitoses or cellular atypia. In a small part, the tumor cells shaped intercalated duct formed follicular structure similar to the thyroid follicle with abundant eosinophilic homogeneous secretion (Figure 2c,d). In addition, in the marginal area of the tumor, the tumor exhibited a higher-grade component characterized by solid growth, increasing N/C ratio and

strong cellular atypia such as nuclear chromatin concentration, morphological irregularity (Figure 3a,b). In this way, this tumor was a varied tissue construction such as papillary, solid, follicular-like structure. Bone destruction and lymphovascular invasion were also identified (Figure 3c,d).

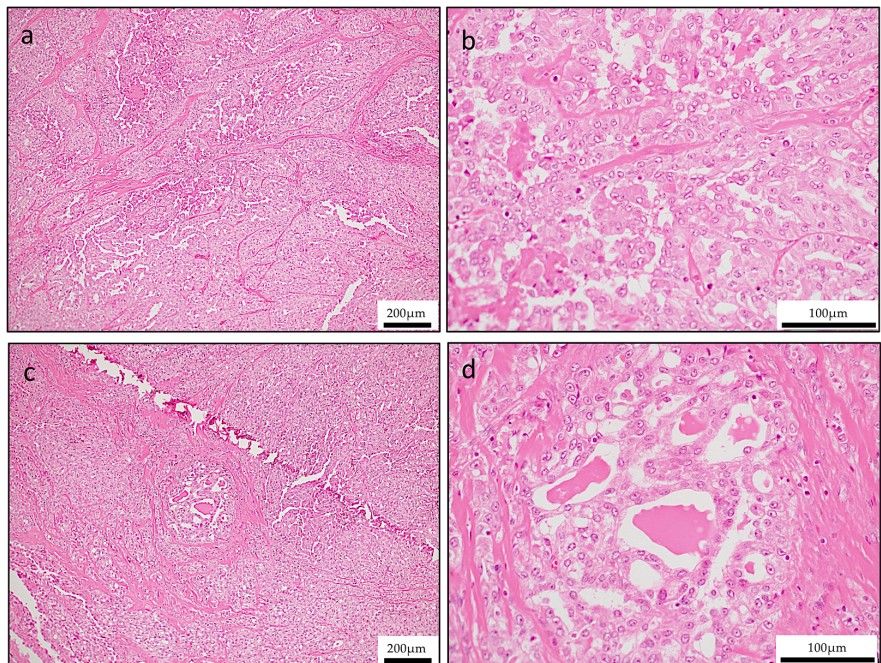

**Figure 2.** Hematoxylin & Eosin (H&E) staining of tumor revealed various cell populations in primary lesion. (**a**,**b**) Lobular or papillary growth pattern (a; low-magnification, b; high-magnification. (**c**,**d**) Solid and follicular growth pattern (c; low-magnification, d; high-magnification).

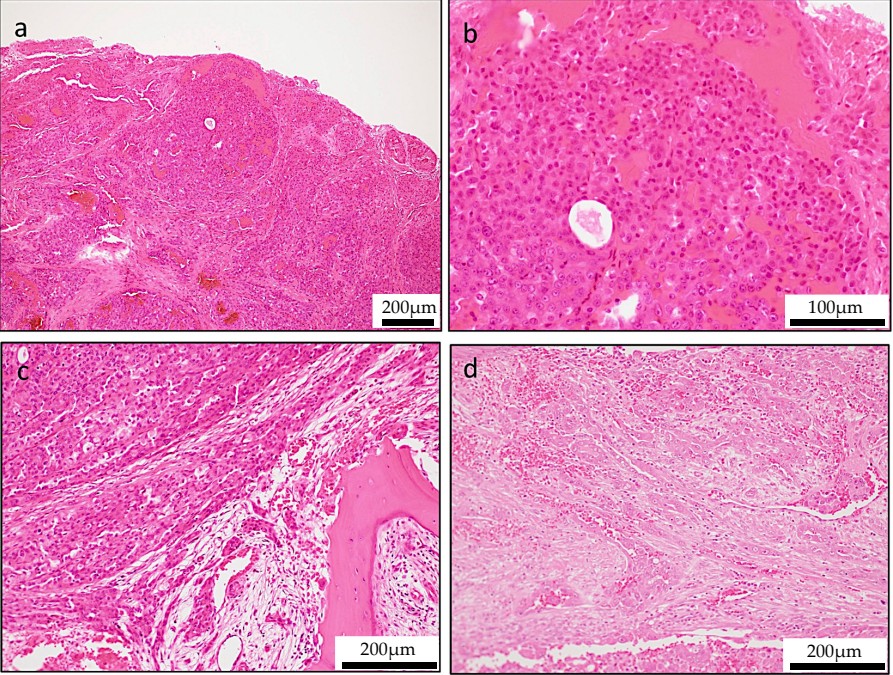

**Figure 3.** Unusual histologic features of secretory carcinoma (SC). (**a**,**b**) Solid growth pattern with high-grade cellular features (a; low-magnification, b; high-magnification,). (**c**) Bone destruction feature. (**d**) Lymphovascular invasion findings.

Immunohistochemical stain showed that the tumor cells were positive S-100 (Figure 4a), vimentin (Figure 4b), AE1/3 (broad spectrum cytokeratin's) (Figure 4c), epithelial membrane antigen (EMA) (Figure 4d), mammaglobin (Figure 4e), whereas calponin, α-smooth muscle actin (SMA), p63, and GFAP (glial fibrillary acidic protein) were negative (Figure 4f).

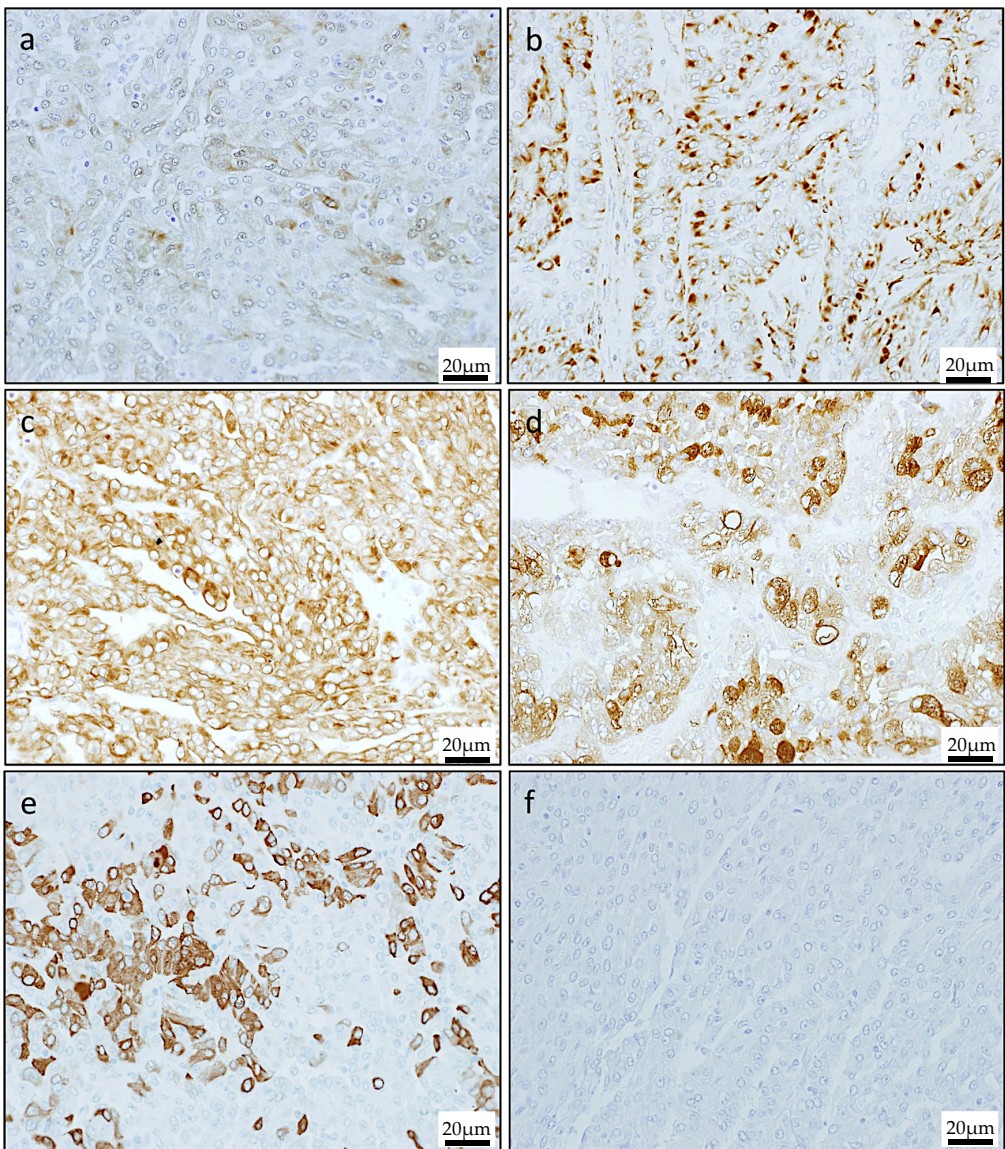

**Figure 4.** Immunohistochemical staining of the primary lesion. The cells were strongly positive for S-100 (**a**), vimentin (**b**), AE1/3 (**c**), epithelial membrane antigen (EMA) (**d**), and mammaglobin (**e**), but negative for GCDFP-15 (**f**).

In special stains, mucicarmine and diastase periodic acid-Schiff (d-PAS) were positive at a pale eosinophilic mucous-like substance between the papillary epithelium, and zymogen granules that was characteristic of acinic cell adenocarcinoma were observed (Figure 5a,b).

Ki-67 proliferative index ranges were a low level (average, 6.1%; range, 4.1% to 7.3%) in the entire lesion. However, in sites with solid growth of tumor margin, Ki-67 showed a high positive rate (average, 38.1%; range, 32.4% to 43.9%). These morphologic features were consistent with high grade transformation (Figure 5c,d).

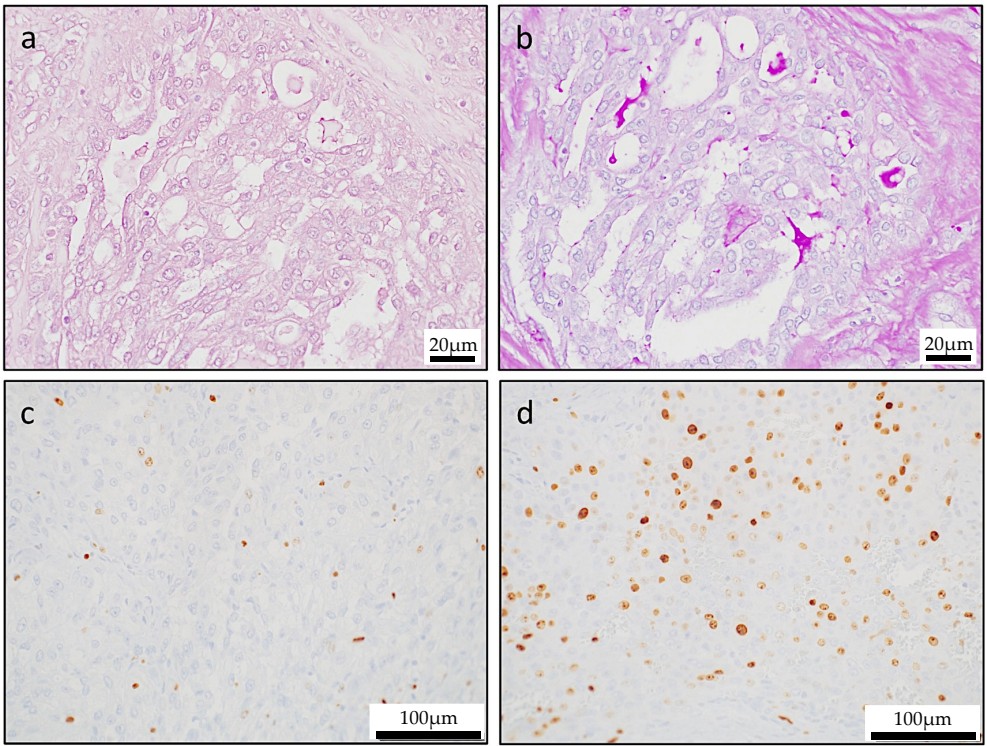

**Figure 5.** Special stains and Ki-67 labeling index. (**a**) Mucicarmine staining revealed mucinous cells in the tumor. (**b**) Diastase-digestion periodic acid-Schiff (PAS) staining revealed diastase-resistant PAS-positive globules. In the primary lesion, the Ki-67 labeling index was low in the papillary area (**c**), but the Ki-67 labeling index was high in the solid area (**d**).

### 2.2. Pathological Findings of Metastasis Lesion

The tumor in the metastasis lesion was composed of two contrasting areas (Figure 6a). One component was papillary pattern which was similar to primary lesion findings (Figure 6b). Follicle pattern lesion where eosinophilic secretion filled within microcystic space resembled the histology of mammary gland secretory carcinoma was also observed (Figure 6c). In immunohistochemical stain, mammaglobin and GCDFP-15 were negative in papillary pattern lesion (Figure 6d,g); on the other hand, in follicle pattern lesion, mammaglobin was positive and GCDFP-15 was negative (Figure 6f,h).

### 2.3. Molecular Findings

We considered the diagnosis of the lesion as SC histologically and immunohistochemically, however in both primary and metastasis lesion, the histological findings were diverse and were not typical findings. Therefore, we searched for the *ETV6-NTRK3* fusion gene in order to make a definitive diagnosis. *ETV6-NTRK3* fusion transcript by RT-PCR in paraffin-embedded tissue specimens were used to identify SC. Details of the genetic search method were performed as described previously [8] and the detail are described below. RNA from this case was extracted using the RureLink FFPE RNA Isolation Kit (ThermoFisher, Tokyo, Japan). Synthesis of complementary DNA (cDNA) was performed using the Transcriptor First Strand cDNA Synthesis Kit (RNA input 1 μg) (Roche Diagnostics, Mannheim, Germany). All procedures were performed according to the manufacturer's protocols. Detection of 110 bp fragments of *ETV6-NTRK3* fusion transcripts was performed by RT-PCR. We added 2 microliters of cDNA to a reaction mixture containing 12.5 μL of Hot Star Taq PCR Master Mix (NEW ENGLAND BioLabs, Inc., Ipswich, MA, USA), 10 pmol of each primer TRKC1059 complementary to NTRK3 with sequence (5′-CAGTTCTCGCTT CAGCACGATG-3′) (ThermoFisher, Tokyo, Japan) and TEL971 complementary to ETV6 with sequence (5′-ACCACATCATGGTCTCTGTCTCCC-3′) (ThermoFisher, Tokyo, Japan) and distilled water up to 25 μL. The amplification program comprised of denaturation at

95 °C for 14 min and 45 cycles of denaturation at 95 °C for 1 min, annealing at 65 °C for 1 min and extension at 72 °C for 1 min. The program was finished by incubation at 72 °C for 7 min. Genetic search was performed on the primary tumor (Figure 7a) and metastatic lymph node (in two different histological findings site) (Figure 7b). We diagnosed this lesion as SC because the *ETV6-NTRK3* fusion gene was detected at both the primary and metastatic lesions (Figure 7c).

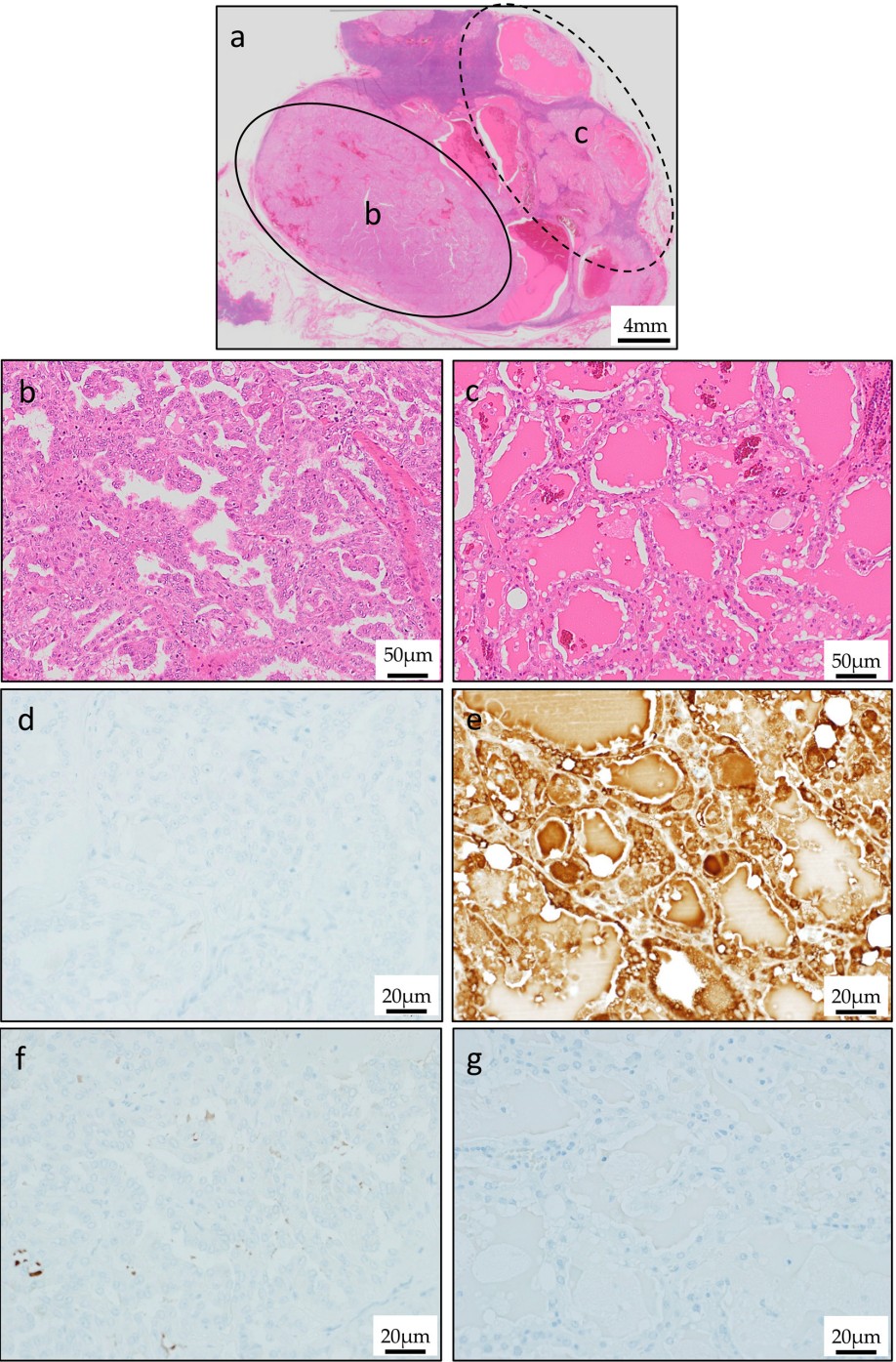

**Figure 6.** Characteristic histologic pattern of metastasis lesion. (**a**) H&E staining of metastasis lesion. (**b**) Papillary appearance with resemblance to primary lesion. (**c**) Follicular and macrocystic appearance with resemblance to thyroid gland colloid. (**d**)Mammaglobin was negative in papillary pattern lesion. (**e**) Mammaglobin was positive in follicular pattern lesion. (**f,g**) GCDFP-15 was negative in both papillary pattern lesion and follicular pattern lesion.

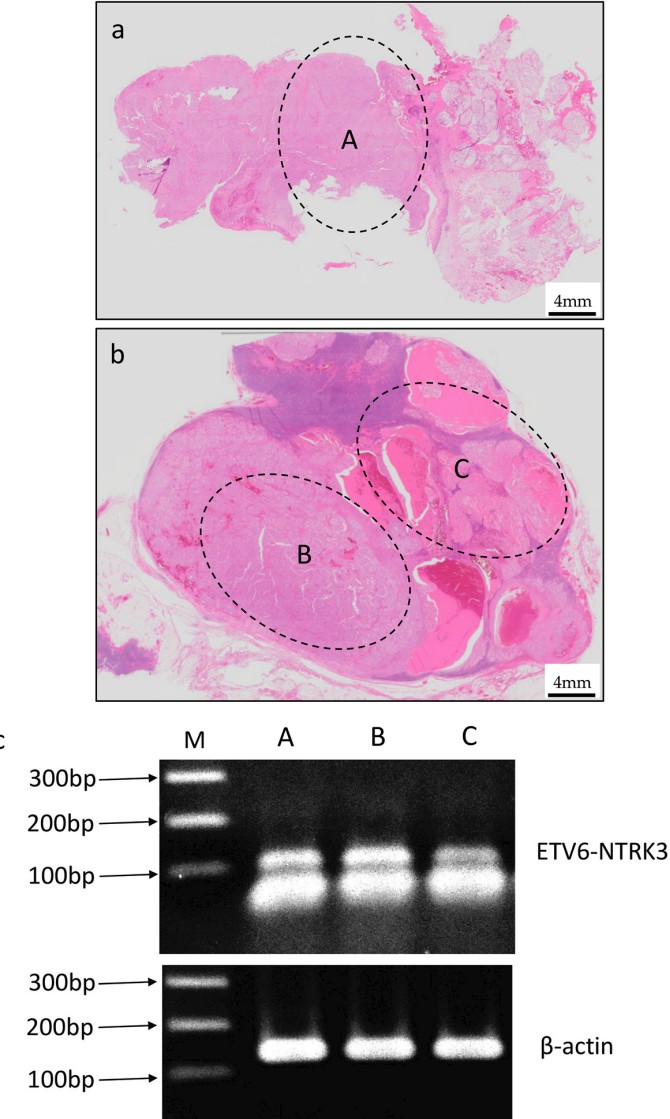

**Figure 7.** Expression of *ETV6-NTRK3* fusion transcript in the primary lesion and the metastasis lesion by RT-PCR. (**a**,**b**) The tissue collection sites of primary and metastasis lesions. (**c**) *ETV6-NTRK3* fusion transcript confirmed by RT-PCR. Line A was primary lesion and Line B and C were metastasis lesions.

## 3. Discussion

Secretory carcinoma of breast that was first reported by McDivitt in 1996 is one of the low histologic grade adenocarcinoma, which is subtype of mammary carcinoma, produces lactational secretions, and shows diverse histological findings [22]. Then, Tognon et al. identified a t(12;15)(p13;q25) resulting in an *ETV6-NTRK3* gene fusion [23]. Hirokawa et al. pointed out the histological proximity between salivary gland acinar cell carcinoma and mammary secretory carcinoma [1], and then in 2010, when Skalova et al. investigated EN gene fusion at 30 salivary carcinomas, EN gene fusion was detected in 16 cases [3]. Therefore, Skalova et al. reported carcinomas resembling mammary gland secretory carcinoma in the salivary gland as mammary analogue secretory carcinoma (MASC). In 2017 WHO classification, secretory carcinoma (SC) from mammary analogue secretory carcinoma (MASC) was classified as secretory carcinoma (SC) [12].

It is reported that MASC accounts for <0.3% of all salivary gland tumors [24], and MASC makes up 4.5% of malignant salivary gland disease processes [25]. In the most recent update, there have been a total of 248 MASC cases reported in the literature, and 68 (24%) cases of MASC reported in minor

salivary glands [4–11,26]. To date, 12 cases of MASC on the hard palate and six on the soft palate have been reported in the literature worldwide [26].

Histologically, SC typically exhibits papillary-cystic, microcystic, tubular or solid architectures, and is composed of uniform cells with bland vesicular nuclei and eosinophilic to vacuolated cytoplasm. At the follicular site, the intercalated ductal tumor grows and takes a structure similar to the thyroid follicles that contain the eosinophilic substances in the small to medium size cavity, and this form is quite similar to the histology of the mammary gland secretory cancer. Tumor cells have oval nucleus and have vacuolated cytoplasm and foamy secretions. Cytological atypia is not strong and it is rare to admit mitosis images.

Immunohistochemically, S-100 protein [3,13,14,17,27] and vimentin [3,4,8,10,14,15,27–38] were strongly positive in all SC cases. Although vimentin is not a specific marker, it can be used as an indicator of myoepithelial differentiation. Positive rates of mammaglobin [3,13,14,17,27], STAT5a [3,28], GCDFP-15, CK7, 8,19, HMWK, MUC1, adipophilin etc. were high; on the other hand, calponin, SMA and p63 were often negative [3]. In addition, Urano et al. and Miesbauerova et al. reported that SOX-10 expression was also identified in SCs to support and confirm the diagnosis of SC [38,39]. There are many reports that DOG1 is negative in recent years, and it is said that the usefulness of S-100 protein, mammaglobin, adipophilin, and DOG 1 is high for distinguishing between SC and other salivary gland tumors, but *ETV6-NTRK3* fusion gene search is essential.

SC is typically a low-to-medium grade malignancy with low grade histopathologic features, thus aggressive features are uncommon in SC. However, extracapsular/extraglandular extension and perineural invasion have been reported in several cases [15], and our case showed high-grade transformation findings (HGT) in the edge of primary lesion. In SC, the first case of HGT was reported by Jung et al., the authors noted the histological feature resemble to high-grade differentiated acinic carcinoma [17]. As far as I searched, only 16 cases HGT SC has been reported [7,8,11,17–21]. Histologically, the high-grade transformation area is characterized by solid growth with atypical tumor cells and the high-grade transformation area with comedo necrosis adjacent to follicular structure site tend to demonstrate a more aggressive nature and ultimately a poor prognosis [6]. In addition, HGT cases had lymph node metastasis and were reported to affect prognosis. However, there are still few HGT reported cases, it is considered to be a rare case, and further clinical cases should be accumulated in the future in order to clarify clinical pathology and biological characteristics.

In our case, the diverse histological findings such as papillary, follicular and HGT lesion were observed in both primary and metastasis lesions, thus it was difficult for us to make diagnosis with histological findings alone. However, even in various histological findings, *ETV6-NTRK3* fusion gene was detected at both the primary and metastatic lesions. Therefore, *ETV6-NTRK3* fusion gene search is essential for SC diagnosis. However, for future study, we should perform the detection of *ETV6-NTRK3* gene break by Fluoroscence in situ hybrization (FISH) to confirm the RT-PCR results on all localization. To the best of the authors' knowledge, the present case is the first report that *ETV6-NTRK3* fusion gene is detected at any sites of the various histological findings and HGT lesion of SC.

**Author Contributions:** Conceptualization, K.T. and K.N.; methodology, K.N.; investigation, K.T., H.K., H.O., S.Y., M.W.O., and S.Q.; resources, K.U. and K.M.; writing—original draft preparation, K.T., K.N. and H.N. All authors have read and agree to the published version of the manuscript.

**Funding:** This study was jointly funded by the Japan Society for Promotion of Science (JSPS) KAKENHI Grant-in-Aid for Scientific Research (16K11722, 18K09789, 18K17224, 19K19159, 19K24094 and 19K19160).

**Conflicts of Interest:** The authors declare no conflict of interest.

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
