# Peer review of "Secretory Carcinoma of Salivary Gland with High-Grade Histology Arising in Hard Palate: A Case Report"

_reports, doi:10.3390/reports3020006_

Round 1
Reviewer 1 Report
The authors describe here a case of secretory carcinoma (SC) with high grade transformation.
To my point of view this article do not add any novelty. The quality of images is low and need to be improved. Some scales are missing (fig 1A). Other scales are not readable (too small). A material and methods part is missing notably to explain how the authors detect the ETV6-NTRK3 fusion transcript? Why do we see two bands?
There is no real review of the literature as mentioned in the abstact.
Author Response
- To my point of view this article do not add any novelty.→This case report novelty is the first report that ETV6-NTRK3 fusion gene is detected at any sites of the various histological findings and HGT lesion of SC. And we have added the sentence in Line 180-181
- The quality of images is low and need to be improved.→We have improved the quality of images.
- Some scales are missing (fig 1A). Other scales are not readable (too small).→We have modified the scale bars in figures.
- A material and methods part is missing notably to explain how the authors detect the ETV6-NTRK3 fusion transcript?→We have added the method of detection of the ETV6-NTRK3 fusion transcript in Line122-123.
- Why do we see two bands? →These band was ladder band of RNA degradation products.
- There is no real review of the literature as mentioned in the abstact. →We have already mentioned the immunohistochemistry and high grade transformation of SC using references. And we have added the sentence about HGT in Line 169-171.
Reviewer 2 Report
The authors present an interesting case report of a rare tumor even in more rare localization with nice illustrative figures. It is a nice teaching case.
The immunohistochemical profile and the discussion of IHC utility should be updated as vimentin is non-specific marker, even of myoepithelial differentiation and wide availability. SOX10 should be added and discussed as well as DOG1/p63 negativity - see for example recent sereis by Miesbauerova in APMIS.
There are few typos to be corrected (line 30, 132 just to mention few).
Author Response
- The immunohistochemical profile and the discussion of IHC utility should be updated as vimentin is non-specific marker, even of myoepithelial differentiation and wide availability. SOX10 should be added and discussed as well as DOG1/p63 negativity - see for example recent sereis by Miesbauerova in APMIS.→We have added the comment and the references about vimentin in Line 154-156, and we have discussed and added the references about SOX-10 in Line 158-160.
- There are few typos to be corrected (line 30, 132 just to mention few).→We have corrected some typos.
Round 2
Reviewer 1 Report
The authors did not answer all my comments.
They added one reference about RT-PCR but did not described the method. In fact, i would be useful to understand why herein they have two bands, but only one band is visible in Majewska H et al paper.
Moreover, if the second band is a"ladder band of RNA degradation products." why don't we see it on the gel with beta-actin?
FISH with ETV6 or NTRK3 break apart probe may be performed on all localisations to confirm the RT-PCR results.
Fig7 legend: Replace PCR by RT-PCR
Author Response
They added one reference about RT-PCR but did not described the method. In fact, i would be useful to understand why herein they have two bands, but only one band is visible in Majewska H et al paper.
Moreover, if the second band is a"ladder band of RNA degradation products." why don't we see it on the gel with beta-actin?
→We have added the mention of the method in detail in Line 123-135.
→It was possible that non-specific amplification of RNA has increased because the annealing temperature was a little low. However, we detected the ETV6-NTRK3 band.
FISH with ETV6 or NTRK3 break apart probe may be performed on all localisations to confirm the RT-PCR results.
→We agree that FISH may be performed on all localization to confirm to the RT-PCR, we have included your point as a consideration for future study (Line 192-193). Thank you for the suggestion.
Fig7 legend: Replace PCR by RT-PCR
→We have modified Fig7 legend.
Again, thank you for giving us the opportunity to strengthen our manuscript with your valuable comments and queries. We have worked hard to incorporate your feedback and hope that these revisions persuade you to accept our submission.
Round 3
Reviewer 1 Report
The authors answered my comments.
Authors should add the company from which they obtained the primers.
Author Response
Authors should add the company from which they obtained the primers.
→We have added the company name in Line 131-133.